# Evaluation of Conserved RNA Secondary Structures within and between Geographic Lineages of Zika Virus

**DOI:** 10.3390/life11040344

**Published:** 2021-04-14

**Authors:** Kevin Nicolas Calderon, Johan Fabian Galindo, Clara Isabel Bermudez-Santana

**Affiliations:** 1Departamento de Biología, Universidad Nacional de Colombia, Bogotá 111321, Colombia; kncalderong@unal.edu.co; 2Departamento de Química, Universidad Nacional de Colombia, Bogotá 111321, Colombia; jfgalindoc@unal.edu.co

**Keywords:** Zika virus, phylogenomics, viral genomic variability, conserved RNA structures

## Abstract

Zika virus (ZIKV), without a vaccine or an effective treatment approved to date, has globally spread in the last century. The infection caused by ZIKV in humans has changed progressively from mild to subclinical in recent years, causing epidemics with greater infectivity, tropism towards new tissues and other related symptoms as a product of various emergent ZIKV–host cell interactions. However, it is still unknown why or how the RNA genome structure impacts those interactions in differential evolutionary origin strains. Moreover, the genomic comparison of ZIKV strains from the sequence-based phylogenetic analysis is well known, but differences from RNA structure comparisons have barely been studied. Thus, in order to understand the RNA genome variability of lineages of various geographic distributions better, 410 complete genomes in a phylogenomic scanning were used to study the conservation of structured RNAs. Our results show the contemporary landscape of conserved structured regions with unique conserved structured regions in clades or in lineages within circulating ZIKV strains. We propose these structures as candidates for further experimental validation to establish their potential role in vital functions of the viral cycle of ZIKV and their possible associations with the singularities of different outbreaks that lead to ZIKV populations to acquire nucleotide substitutions, which is evidence of the local structure genome differentiation.

## 1. Introduction

Zika virus (ZIKV) was first identified in Rhesus monkeys in the Zika forests of Uganda in 1947. Its spread from Africa reached Southeast Asia, limiting its associated symptoms to feverish symptoms, conjunctivitis and joint pain during the 20th century [1]. At the beginning of the XXI century, outbreaks of the virus began in countries of the island complex of Oceania, where new symptoms were associated, such as Guillain-Barré Syndrome (GBS), an autoimmune disease in which the immune system attacks the nervous system of the affected person [2]. Since its dispersion in America in 2015, the virus infection started to be associated with congenital fetal microcephaly and neurological damage caused by the virus’s vertical transfer between the mother and fetus [3,4]. Declared as a public health emergency by World Health Organization (WHO) in 2016 and due to the absence of a vaccine in preventive terms or specific treatment, ZIKV is currently among the most significant concerns of health systems in tropical countries [5], where its transmission occurs mainly by vector mosquitoes of the *Aedes* genus [6].

ZIKV is a single-stranded positive-sense RNA arbovirus within the *Flavivirus* genus [7], the genus to which other known viruses of public health importance belong, such as Dengue (DENV), yellow fever (YFV), or West Nile Virus (WNV) [8]. The genome length is close to 10.8 kb, and is composed of two untranslated regions (UTR) located at the 5’ and 3’ ends, of 106 and 428 nt in length, respectively [9], and a coding region (CDS) of 10.3 kb. Capsid (C), Membrane (M), and Envelope (E) (CDS) codes for three structural proteins, and for seven non-structural proteins (NS1, NS2A, NS2B, NS3, NS4A, NS4B, and NS5), necessary to complete its viral replicative cycle [8].

The ZIKV genome folds in a secondary RNA structure shape like RNA-type molecules which also occurs in other RNA viruses, where these structures perform pivotal functions during the viral cycle, such as regulating translation, promoting replication and evading host cell antiviral responses [10,11,12]. For example, the Flavivirus genus is known to exploit the structures located at the 3’ UTRs to produce non-coding RNAs (ncRNAs) named flaviviral subgenomic RNAs (sfRNAs). sfRNAs, in particular, have been associated with antiviral response evasion by negatively affecting the immune response mediated by interferon type 1 (IFN-I) of the host cell [13,14]. The viral replication in Flaviviruses is also initiated by changing their linear genome into a circular genome through the interaction of the RNA structures located at the two ends of the strand [15]. In particular, it has been reported that the structures of the 5’ UTR region in ZIKV regulate the initial viral translation by promoting the placement of a CAP at the 5’ end and mimicking the mRNA of the affected cells [16]. Additionally, many RNA viruses potentially encode precursor structures of microRNAs processed by canonical and non-canonical pathways in the host cell [17,18]. These microRNAs can affect the host cell’s metabolism or regulate antiviral response from the genes’ expression [19,20,21,22].

However, patterns of RNA secondary structures in viruses with RNA genomes, despite their importance, have been poorly studied [23] in a comparative genomics. Here, we present an extensive survey to search for ZIKV-conserved genomic subregions restricted to specific continental geographical origins. We found that some structures are conserved by each geographic location or even by lineage classification. We detected critical subregions in the ZIKV genome with our strategy and targeted future studies to establish their function on the viral cycle and its infective capacity [12].

## 2. Materials and Methods

A graphical scheme of rational design of materials, methods and results can be seen in Appendix A.

### 2.1. Genomic Data Source 

An in-depth search for complete Zika virus genomes was performed in NCBI and VipR databases (search date: July 2020) to retrieve a total of 1023 genomes [24,25]. A local database was built with all those sequences. Each sequence was compared against the local database, and redundant sequences were filtered using BLASTn [26] by deleting the sequences with more than one hit. Parameters for BLASTn search were as follows: word size = 28, gap existence = 1, gap extension = 1, match = 1 and mismatch = −2. In the same way, sequences that exceeded 0.05% of unassigned nucleotides "N" were removed using Bioperl tools v1.7.7 [27]. To work with equivalent lengths of sequence size, we used R v4.0.2 software to identify and remove irregularly short sequences (lower outliers of a boxplot of all sequences length) [28]. Therefore, we obtained a final dataset with 410 complete ZIKV genomes to conduct downstream analysis.

### 2.2. Genomic Alignments According to the Geographical Origin

All multiple genomic alignments were performed by Clustal Omega v1.2.4 [29], setting two iterations per alignment and using the other parameters by default. The first alignment included all sequences (global context), and their extremes were trimmed as long as the gap content was greater than 33%, using UNIPRO-Ugene v33.0 software [30]. Subsequently, the sequences corresponding to different continental geographical origins (Africa, Asia, Oceania and America) were filtered from this alignment. Once the group of sequences was set, they were de-aligned and re-aligned by specific geographic regions.

### 2.3. Phylogenomics Analysis

The phylogenetic analysis was based on the whole genomic sequences (including both CDS and UTR regions), and the following R packages were used to evaluate genomic sequence relationships: APE, seqinr and Phangorn [31,32,33]. A distance matrix was built using the dist.alignment function based on the square root of pairwise distances from multiple sequence alignments. A Neighbor-Joining (NJ) phylogenetic tree was performed, considering 1000 bootstrap replications and the yellow fever virus (NC_002031.1) as the outgroup. For genomic alignments, the best-fit model of nucleotide substitution (GTR) was selected under the Akaike information criteria using phymltest in R. Maximum-likelihood trees, with aLTR statistics for the support of internal nodes, and with NJ tree with bootstrap support as input, were inferred using PhyML v3.0.1 [34]. Phylogenetic trees were plotted and visualized and edited with R.

Finally, to describe the variability and conservation of sequences, percentages of pairwise identity and the number of identical sites per nucleotide columns of the alignment were calculated using Geneious Prime v2020.1 [35]. From the phylogenetic relationships obtained from the first alignment (global context), subdivisions of clades that were contained within the continents with the highest number of sequences (Asia and America) were proposed: Asia continental, Southeast Asia, Brazil, Colombia, Mexico and Caribbean clades (Table A1). Thus, we obtained eleven groups of sequences that allowed us to perform comparative analysis, at different scales, within and between geographic lineages.

### 2.4. Prediction of Conserved Secondary Structures

In order to analyze the geographic similarity of the RNA conserved structures, RNAz v2.1.1 software [36] was employed. An experimental design was carried out testing six different combinations of two factors: window size (150, 120 and 100 nt) and sliding window size (20 and 40nt), in order to set the screening parameters and minimize the rate of false-positive prediction. For each parameter combination, 100 randomizations of each alignment were performed using the RNAz script, RandomAlign.pl, to determine false positives (FPrandom) and positive detections of the original alignment (PNative). The relationship between these two indices was established as a quantitative quality criterion (FPrandom/Pnative). The lower the numerical value of this relationship, the higher the specificity and the higher the sensitivity. Based on this, a two-way analysis of variance (ANOVA) was performed (homoscedasticity and normality assumptions were verified), and Tukey plots and Boxplots were made to guide a statistical decision regarding the selection of the sliding window size. Once the best combination of the sliding window was set, we followed the pipeline of the RNAz program, described in Gruber et al. (2010) [29], using the following as a filter: P > 0.9 and Z value < −2; as well as the "no-reference" and "both-strands’’ parameters. Finally, the genomic positions of conserved secondary structures were plotted using ggplot2 v3.3.3 [37]. The index produced by RNAz in HTML was used to graphically extract the most representative RNA secondary structures of each position, considering the following as selection criteria: the Z Value, SCI, SVM decision value and MFE. These representative structures, specifically, their consensus structure sequences, were compared against the families for the Zika virus reported in the RFAM [38], using BLASTN and employing the same parameters described previously. The genomic positions of the structures laying on the envelope coding region were confirmed using a BLASTx search between the consensus sequence calculated by RNAz and the reference sequence of the envelope protein (ANC90422.1) from NCBI. The parameters used were as follows: Identity = 100%, Query cover = 98% and E-value = 6 × 10^−27^.

Finally, as a statistical support of the secondary RNA structures, two different randomization alignment routes were performed: general (to each alignment) and specific (to each window). In the general randomization, 100 randomized alignments from each geographic or subgeographic region were obtained (n = 53,100 windows per alignment). These were analyzed entirely in RNAz, maintaining the size of the sliding window, Z value and P-value parameters. The relative false positive detection rate (FP) and the percentage of specificity (% Specificity) were determined. The specific randomization (200 windows, n = 200) per each of the windows detected as structured in the original alignment was generated, and they were analyzed with the same RNAz parameters mentioned above, removing from the analysis those windows that had a rate of false positives greater than 0.05 (% FP > 0.05).

## 3. Results

### 3.1. Phylogenomics Analysis

The NJ approach shows the diverse relationships between the viral genomes and the continents (Figure 1A). A group of African genomes is observed in the cladogram’s basal position, representing their ancestral status with respect to other continents. Likewise, Asia and America form two large uniform clades, with solid support in their basal branches (bootstrap > 0.9); and they agree in the order of ancestry: first Asia and then America. Additionally, the derived branches that do not have adequate support (bootstrap < 0.7) coincide in having very short branches (<0.01), and their pairwise identity distances are minimal; thus, the pairs of sequences in these branches are very similar sequences. Finally, Oceania does not present a concise continental separation (bootstrap < 0.7), and it is included within the American clade.

The circular design of the cladogram was changed to a traditional design (Figure 1B) to provide a better detail of the continental subgrouping. The distances between paired sequences are not taken into account in Figure 1B, but the groups of sequences and their support are better appreciated. We can see that Asia is divided into Continental Asia and Southeast Asia, with good branching support (bootstrap > 0.9). In the same way, America has been split into at least four subgroups (or clades). Colombia and Mexico clades are well supported in their basal branching (bootstrap > 0.9), and they are also composed of countries from Northern South America and Central America, respectively (detailed list of the set of countries in Table A1). Most of the Caribbean countries are in another clade, which is well supported in a single group (bootstrap > 0.9) and contains the island countries from the Caribbean and the coast of the United States. However, Puerto Rico is separated from this homogeneous group and is a particular case, even though it is at the same geographical position as Figureother Caribbean countries. Finally, we can see a Brazilian clade, which permeates all the other groups in the Americas in the cladogram, reflecting their condition as the original location of the outbreak in the Americas, since the spread was derived from there to the other American countries of America [39]. This representation agrees and explains the scarce differentiation found in the branches with low support and short distance in Figure 1A.

Regarding Maximum Likelihood (ML) phylogenetic trees (Figure 2), these are consistent with the topology generated by NJ. Similar clades to the NJ methodology are observed, i.e., Caribbean, Colombia, Mexico, Brazil, Southeast Asia, Continental Asia and African country blocks (Figure 2B). Therefore, these groupings are independent of possible biases related to the dendrogram graphing methodology. The main difference between both methods lies in that ML separates the sequences from Oceania, and the shorter pairwise distances of ML branches (Figure 2A) make its visualization more difficult. 

In the descriptive analysis of sequence variability, summarized in Table 1, we observe a high sequence conservation level. The percentage of pairwise identity, taking all genomes as a set, is 98.29%, and in none of the proposed clades was less than 99%. This result reflects the high degree of conservation between the sequences. Africa stands out as the region whose sequences show the least similarity among them (94.11%). Despite the high degree of similarity observed between sequences, the number and percentage of nucleotide columns, which are identical in each alignment, are always lower than their percentage of pairwise identity. Finally, the median of the sequences was 10,729 nt, which is close to the length of the ZIKV reference sequences (10.8 kb).

### 3.2. Prediction of Conserved Secondary Structures

From the ANOVA analysis, significant differences in the choice of window size were found (to select the initial parameters of RNAz, window and slide size; Appendix A), but not in terms of sliding size; the interaction between these two parameters is not significant (Appendix A). The Tukey test and the boxplots suggest that we can choose a window size of 150 or 100 nucleotides. The 150-nucleotide size allows the possibility of detecting RNA secondary structures in their complete form and shows hairpin-like structures, which can be targeted by Dicer or any other cytoplasmic microprocessor. Therefore, 150 window and 20 sliding window sizes were the selected parameters.

In evaluating conserved RNA secondary structured regions in the geographical inter-lineage comparison (Figure 3), the globally conserved structured regions are only those of the initial and final parts of the sequences (5 ’and 3’, respectively). These accomplish crucial functions throughout the *Flavivirus* genus, and their presence reflects a positive control in the detection methodology of structured areas for the viral genome (Figure 3). Additionally, African sequences present a unique pattern of structured regions, containing lineage-specific structures at positions 2.8 and 9.6 kb (Figure 3B). On the other hand, Asia, Oceania, and America share four structured regions at places 1.1, 4.5, 7.1 and 8.5 kb (Figure 3C–E). Similarly, Oceania and America share a structured region at the 3.1 kb position (Figure 3D,E). Finally, Oceania has a particular structured area at position 5.2 kb (Figure 3D). This graph allows us to appreciate three types of patterns: (1) there are conserved structured regions in all sequences; (2) there are unique conserved structured regions based on the particular geographic lineages; (3) patterns in the geographic lineage groups are formed because they share certain conserved structured regions.

In the evaluation within geographical lineages of the conserved secondary RNA regions, the Caribbean clade is the only one presenting two structured zones, which differs from all the other American subregions at positions 5.2 and 9.2 kb (Figure 4A). It is worth clarifying that the double points generated in the position close to 4 kb in the American continent and in the Brazil clade subregion are the consequence of a discontinuity in the sliding window of the RNAz pipeline, and they do not represent a different structured region. Additionally, the structured position of Southeast Asia at position 5.7 kb is the only one that is different from other subregions of the Asian continent (Figure 4B). Finally, the same case of the double points, previously mentioned, occurs in the position of 4.5 kb between Asia and Southeast Asia. In general, there is little variation in terms of the presence–absence of structured regions at intra-geographical level regions.

In the statistical validation by complete alignments, we obtained a false-positive detection rate lower than 5% (FP < 0.05), and a specificity index higher than 95% in all cases; therefore, the filters used to run the pipeline of RNAz (Z value < −2, P > 0.9) were effective in selecting information of the detected structures, and the results were statistically significant (Appendix A). In the other approach of statistical validation, for each of the structured windows obtained from RNAz, cited in Appendix A, a total of 30 windows (FP > 0.05) of the analysis were removed. Window number 32 of the Colombia_Cl caught our attention, which was the only one that represented the removal of an entire structured locus.

The results of the most representative secondary RNA structures at the inter-geographical region level are found in Figure 5, highlighting the one found in the envelope region (Figure 5B), since it has experimental validation [23]. The representative structures of the intra-geographical regions are shown in Figure 6.

On the other hand, RNA structures that overlapped with RNA models from RFAM are shown in Table 2. However, the Flavi_CRE structure was not detected, because its presence occurs at the end of the 3 ’UTR genomic region (~10,697 nt); therefore, the alignment quality clipping process, mentioned in the methodology, could have limited its structural detection.

## 4. Discussion

The findings here reported in both phylogenetic trees agree with the historical records of Zika virus outbreaks. The sequences from Africa are basal, as a viral origin, followed by Asia and last America [6]. According to the ML analysis, the location of the sequences from Oceania, as a sister group to America, agrees with it being the place of viral origin introduced to America [39]. Otherwise, the inclusion of Oceania in the American clade, by the NJ method with low support, could suggest a difficulty in the tree resolution due to the high homology between variants from both continents. Moreover, our phylogenetic trees agree with the one generated by Metzky et al. (2017), in which Brazil is located as the geographical origin of the viral breakout in America [5]. In the basal part of the American clade, the short branches, despite having a low phylogenetic signal, agree with the sequences from Brazil as the origin of the outbreak. These sequences are probably dispersed throughout the continent, since they were still very similar to each other. This pattern is associated with a rapidly spreading viral outbreak by the introduction of a new virus to a population without a history of immune memory to the same virus [5].

Additionally, the grouped sequences in the clades of both types of cladograms reflect the establishment of individual viral genotypes in geographically delimited regions, regardless of the methodology used. We always found the Colombia clade, the Caribbean clade, the Mexico clade, Asia Southeast, Asia Continental, and Africa to have good support in the NJ analysis. These groups agree with other phylogenetic trees reported in the literature with a set of sequences previously reported [5,40].

The global and variants from specific geographic regions full-length sequences were also analyzed using comparative genomics to detect conserved secondary structures to show the contemporary landscape of conserved structured regions with unique conserved patterns. The high degree of homology seen between the Zika viral sequences from distant geographic regions can be contextualized with similar features found in other flaviviruses, for instance, the similarity within Dengue serotypes, where its most variable fragment, the 3 ’UTR region, reaches 97% of pairwise identity [41]. Therefore, finding 98% global identity and 99% at the regional level is not an unusually high value and, indeed, it suggests selective purifying pressures on the Zika virus genome. This is feasible because, in the CDS region, its entire length encodes proteins, which are essential for evading the host’s immune responses and completing their viral replication. Thus, the accumulation of drastic changes in its genome can affect the viral viability [42]. Other authors have suggested a purifying selection in viruses that handle a complete viral cycle inside humans. The majority of viral genomes used in phylogenomics studies come from clinical samples, making it more difficult to uncover the virus´ true diversity. If viral genomes were sampled directly from ZIKV circulating in wild mosquitoe vectors, a greater diversity is to be expected [41]. Indeed, higher variability in the WNV virus found in vector insects has already been reported, whose vector, *Culex spp*, has a greater vector-specific viral diversity in contrast to the variety found in the host vertebrate [43].

Regarding the conserved structured regions of RNA in all the Zika virus sequences, at the 5’ and 3’ UTR ends, several of their essential functions have been reported in the *Flavivirus* genus, especially in structures that were also found in the RFAM database [44,45]. Thus, the SLA structure found in the 5’ region is the structure recognized by RNA polymerase (NS5), which is fundamental in viral replication [36]. Additionally, this structure promotes the addition of 5’ CAP during viral RNA synthesis, which is necessary for viral translation by recruiting the eukaryotic eIF4E binding factor and the subsequent recruitment of 48S and 60S ribosomal units [46,47]. Likewise, the DB structure of the 3 ’UTR region has been related to the formation of sfRNA structures and, indeed, a 30 nucleotides deletion in Dengue virus DB1 has generated an attenuated version of the virus, because it becomes particularly susceptible to type 1 interferon, thus highlighting the importance of this structure in the viral cycle. Something similar may happen in the Zika virus [48]. The differences in structured regions between Africa and the rest of the world are possibly related to biotic particularities of the continent; for instance, the transmission in Africa is performed by another vector species: *Aedes africanus*. It is acknowledged that secondary RNA structures are vital factors in flaviviruses for viral replication in their respective disease-transmitting insects [42]. For example, when DENV is cultured in human cells, structures sfRNA1 and 2 are mainly produced, while culturing the same serotype in mosquito cells produces more types of sfRNAs: 1, 2, 3 and 4 [49].

Another example is a WNV mutant which lacks the formation of the sfRNA1 structure, and it does not survive in the intestine of the insect *Culex spp.*, but it survives in its salivary glands; therefore, it is a key structure to complete the viral cycle, from the ingestion of blood to transmission by mosquito saliva [14]. This is an interesting aspect because it has been shown that the Zika virus can replicate in the intestine and salivary glands of *Culex spp.* insects [50]. The virus may be adapting to new vectors, and its outbreaks may have new scopes linked to the distribution of this other type of vector, all mediated by changes and adaptations in their particular RNA structures.

With respect to the structural region of RNA shared among Asia, Oceania, and America in the position close to 1.1 kb, it is found to be intriguing because it has been experimentally validated, in vivo, and its importance in the function of the Zika virus has been reported [23]. The authors found an intra-molecular interaction of the RNA structures present in the 5’UTR region (2–43 nt) and the structures of the envelope coding region (E) (1089–1134 nt), which occurs only in post-epidemic viral strains of Asia, Oceania, and America, but not in Africa. They evaluated four mutants, damaging their RNA structures of the coding position corresponding to the envelope, without altering the encoded protein, and obtained a reduction in viral infectivity. This infectivity was partially restored by reintroducing compensatory mutations to re-form the structure initially found [23]. This structure coincides with the structured region found in the present work, at 1.1 kb, corresponding to the envelope coding region (Figure 5B). Therefore, this pattern analysis of an RNA structure, which has been associated with viral infectivity, allows the possibility that other structured regions found in this work may also have critical functions in the viral cycle of ZIKV (with unique or shared patterns as are shown in Figure 3).

Finally, the RNA structures found in common between America and Oceania might contribute to the genomic particularity present in the outbreaks, where the tissue damage and viral infectivity were superior in contrast to Asian and African lineages in experimental studies with mice [51]. Additionally, it is remarkable that fragmenting the alignments in sliding windows generates border effects, where in silico structures may be incomplete in their prediction. However, the structures reported here show a strong signal of being a structured region of the genome and facilitates a later evaluation of the real 3D structure. An example of the relation of the RNA 3D structure’s relation to its function can be observed in pre-microRNAs [52].

## 5. Conclusions

Performing this comparative analysis between Zika virus genomes and their RNA conserved secondary structures allowed the selection of regions and certain specific structures, which are distinguished by their patterns in genomic comparison at the inter-geographical lineage level. In this way, these patterns of structural conservation guided the selection of potential functionally relevant structures in the viral cycle of ZIKV. Further experimental analysis for associating new functions must be performed. In the future, these structures may have the potential to be targeted to negatively affect viral replication [12].

## Figures and Tables

**Figure 1 life-11-00344-f001:**
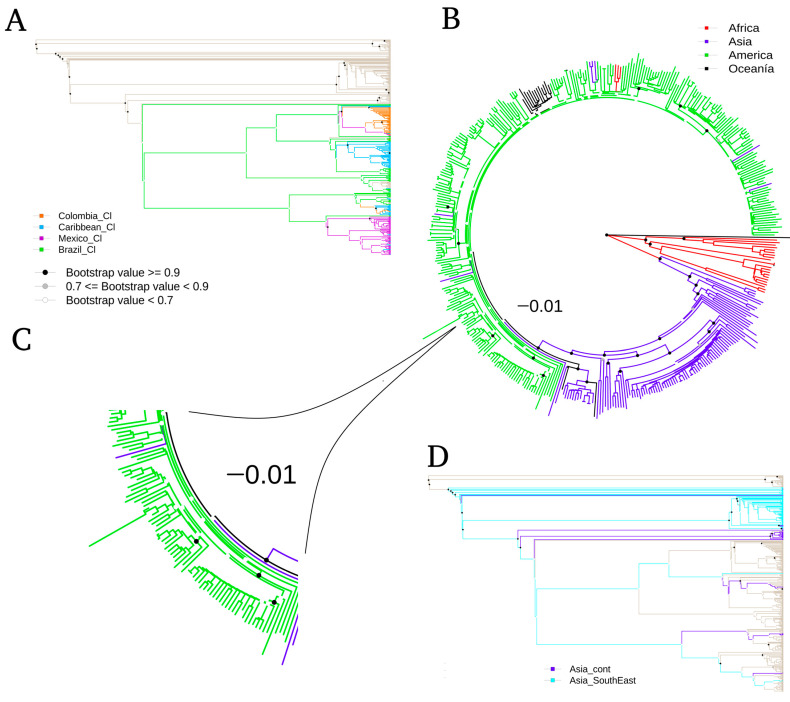
Phylogenetic tree of ZIKV produced by Neighbor-Joining approach. (**A**) Phylogram classic-plot type, intra-geographic linages of America. (**B**) Fan plot type, sequences according to their inter-geographic lineages. (**C**) Zoomed portion of the short branches in the American clade. (**D**) Phylogram classic-plot type, intra-geographic linages of Asia. The intra-geographic plots show relationships among circulating strains from different places in American or in Asia continents. Note that some nodes receive bootstrap values of 1 (100%), indicating strong support for these nodes, whereas other nodes receive much weaker support (e.g., 0.7 (70%)).

**Figure 2 life-11-00344-f002:**
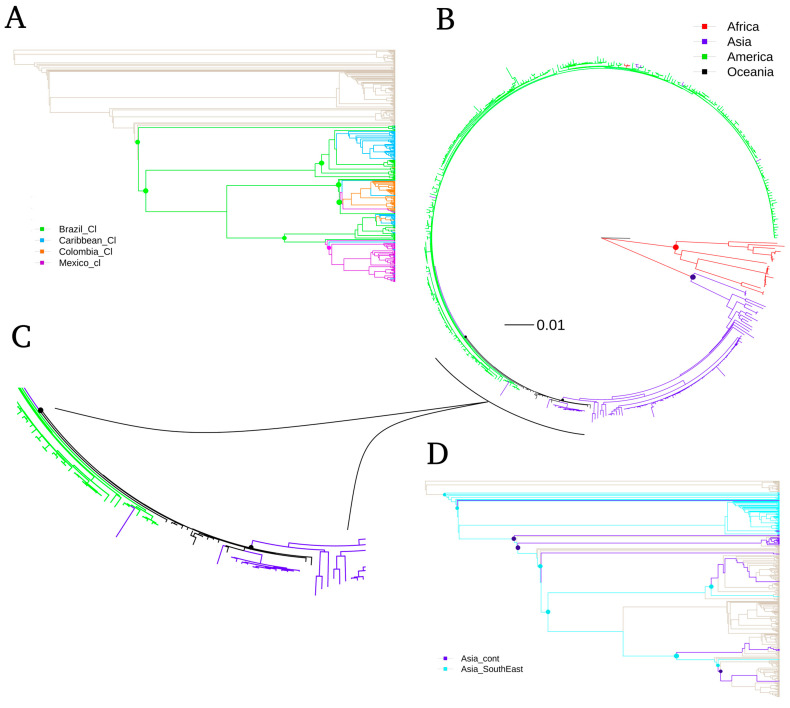
Phylogenetic tree of ZIKV produced by maximum-likelihood approach using the GTR model and the approximate likelihood ratio test (aLTR). (**A**) Phylogram classic-plot type, intra-geographic linages of America. (**B**) Fan plot type, sequences according to their inter-geographic lineages. (**C**) Zoomed portion of the Oceania clade. (**D**) Phylogram classic-plot type, intra-geographic linages of Asia. The intra-geographic plots show relationships among circulating strains from different places in America or in Asia continents. Approximate likelihood-based measures of branch support with phylogentic signal equal to non-zero values are indicated in colored dots and calculated with aLTR statistics.

**Figure 3 life-11-00344-f003:**
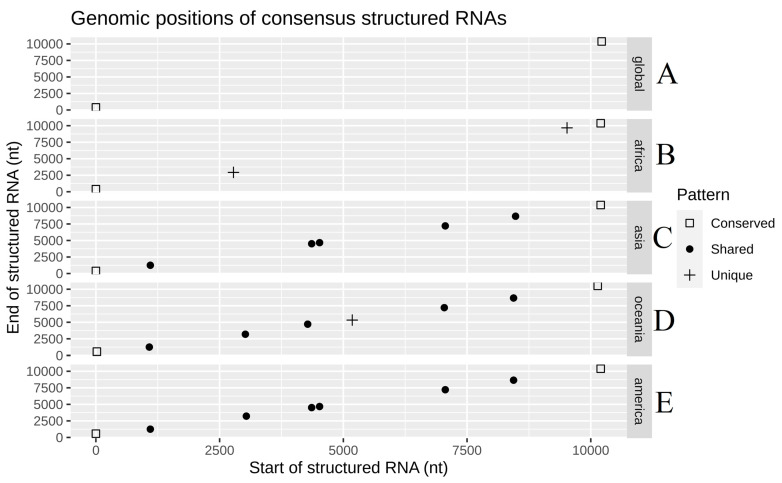
Genomic position and patterns of the structured regions of RNA found in inter-geographic lineages of the Zika virus: (**A**) global, (**B**) Africa, (**C**) Asia, (**D**) Oceania and (**E**) America.

**Figure 4 life-11-00344-f004:**
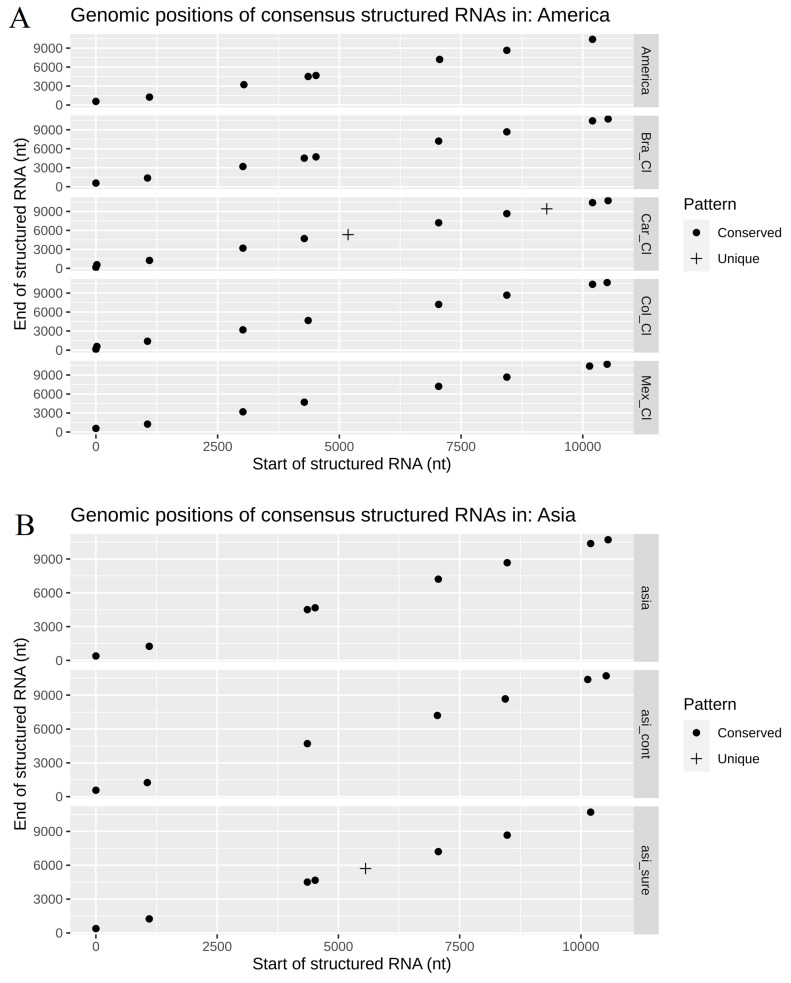
Genomic position and patterns of the structured regions of RNA found in the intra-geographic lineages of the Zika virus: (**A**) America and its respective subregions; (**B**) Asia and its respective subregions.

**Figure 5 life-11-00344-f005:**
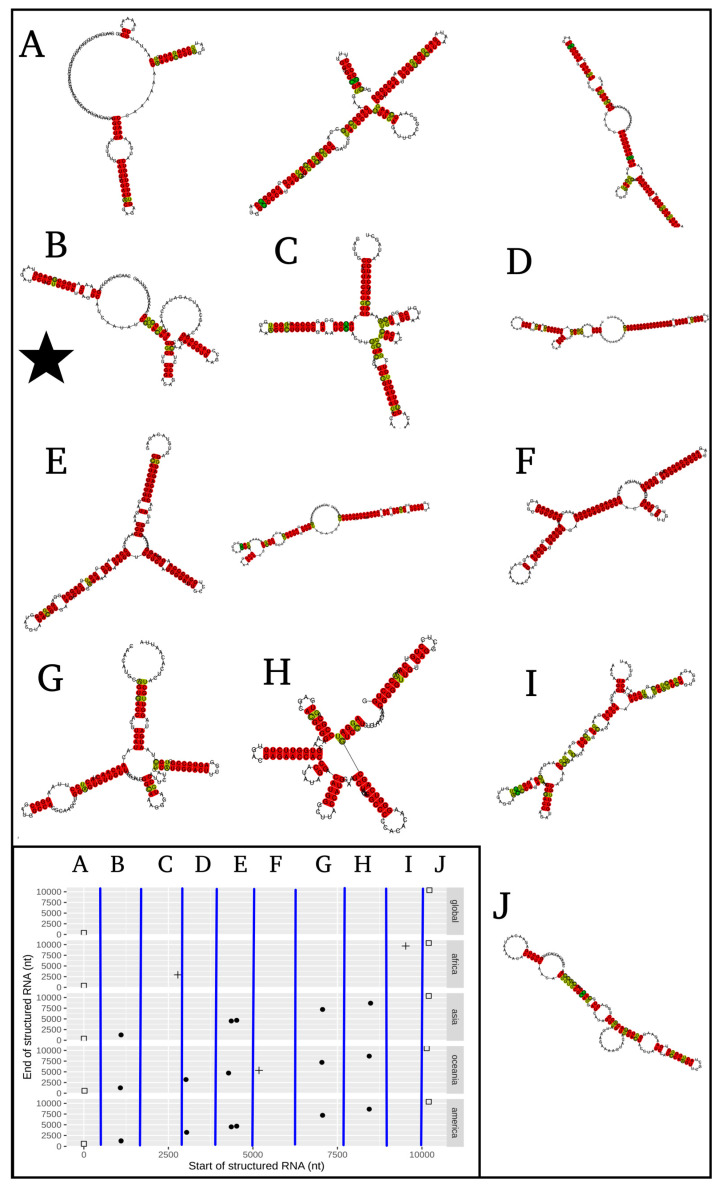
RNA most representative structures of the genomic regions at the inter-geographic lineage level. (**A**–**J**) Each structure, or group of structures, represents the genomic position plotted on Figure 3 which is now embedded in the lower left box. The star indicates the structure related to the envelope coding region reported in [23]; its genome location was traceback by using a BLASTx search, whose parameters are detailed in materials and methods section.

**Figure 6 life-11-00344-f006:**
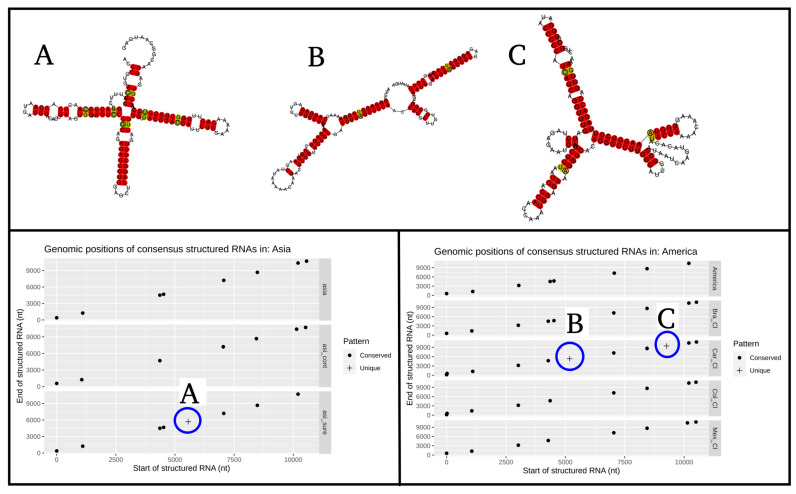
RNA most representative structures of the genomic regions at intra-geographic lineage level comparisons for Asia and America. The blue circles (**A**) correspond to the unique structure found in the south east of Asia, and the blue circles (**B**,**C**) are unique structures found in the Caribbean island clade.

**Table 1 life-11-00344-t001:** Summary table of descriptive data of the sequences analyzed.

		Length		Identical Sites		Mean Pairwise Identity	
Region	Sequences(n)	Median	Range	N° Columns	%	%	SD
Global	410	10,729	10,368–11,119	7205	64.8	98.29	0.031
Africa	24	10,782	10,617–11,119	8917	80.2	94.11	0.037
Asia	106	10,762	10,415–10,808	8970	83.0	99.01	0.009
Oceania	14	10,644	10,585–11,155	11,021	98.8	99.86	0.001
America	266	10,692	10,368–10,864	8973	82.6	99.59	0.001
Brazil_Cl	58	10,752	10,455–10,864	10,288	94.7	99.65	0.001
Colombia_Cl	53	10,659	10,385–10,808	10,375	96.0	99.80	0.002
Mexico_Cl	66	10,696	10,398–10,807	10,191	94.3	99.75	0.001
Caribbean_CL	89	10,727	10,368–10,808	9986	92.4	99.55	0.002

**Table 2 life-11-00344-t002:** Match of the structured windows found in the analysis with the only structures reported in the RFAM for the Zika virus. (p.ident = percent identity, gap.open = gap opening value, q.start = query start, q.end = query end, s.start = subject start, s.end = subject end,/= not applicable).

Annotated Structure	Window	p.ident	Length	Mismatch	gap.open	q.start	q.end	s.start	s.end	E-Value	Bit-Score
Flavivirus DB	16	100	29	0	0	122	150	1	29	3.29 × 10^−11^	49.6
Flavi_SLA	1	100	57	0	0	1	57	17	73	4.48 × 10^−25^	95.7
Flavi_CRE	no hits	/	/	/	/	/	/	/	/	/	/

## Data Availability

Not applicable.

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
