# Peer review of "Evaluation of Conserved RNA Secondary Structures within and between Geographic Lineages of Zika Virus"

_life, 2021, doi:10.3390/life11040344_

Round 1

Reviewer 1 Report

In the presented manuscript Calderon et al. analysed full genome sequences of Zika virus (ZIKV), drawing virus phylogenetic relationship and predicting conserved and unique ZIKV RNA secondary structures. The study was well designed and will be of interest to the scientific community. Below I propose so minor amendments which I believe will improve the manuscript:

  1. Lines 46-49 reads as authors refer to the ZIKV however, however provided references are with regards to RNA viruses in general or flaviviruses, please rephrase. The same occurs in the entire paragraph (lines 46-61). Please be more specific where in the text you refer specifically to ZIKV, where to flaviviruses in general, and where to other RNA viruses.
  2. I assume that the phylogenetic analysis was based on the whole genomic sequences (i.e. as presented in Table 1. Is that correct. One sentence specifying it added to material and methods would be of benefit?
  3. The result section would benefit from a brief introduction to the rationale or design of the experiments.
  4. Fig. 1 and Fig. 2. A larger (i.e., of a higher resolution) fan plot in the main figure and the phylogram being moved to the supplementary material might be more readable. Alternatively, the parts of the fan plot which are important to show could be ‘zoomed into’ using the phylogram.
  5. Line 217: structure of this sentence is a bit confusing, maybe putting “1)” instead of “1.” Or “a)” instead of “1.” would help,
  6. Table 2. Please correct formatting – table should not be split between two pages and caption should be above the table. Table 2 could be moved to supplementary materials.
  7. Fig.5 legend: Please confirm that the lower left box contains Fig. 3.
  8. Fig S2. Could be moved into the main text.
  9. The discussion could benefit from re-phrasing, so it is easier for the reader to understand what the authors are trying to say (i.e., be more straight to the point)

Reviewer 2 Report

The manuscript assesses Zika virus’s structured RNAs, considering 412 complete genomes from various geographic regions, an important human pathogen without an effective treatment.

The methodologies used seem appropriate for the data and the aims of the work, however, they must be more detailed, and several improvements are needed.

There are several details in Figures, Tables and Supplementary materials that need to be corrected and improved, with special attention to the resolution and readability of the information contained in the Figures.

In more details, authors must correct and improve the following aspects throughout the manuscript:

 Abstract:

 Lines 17-18: Please consider rephrasing and add more details. This sentence seems to be too vague.

Introduction:

Line 22: Zika virus instead of zika.

Line 41: Add space “10.8kb” – 10.8 kb and “untranslated region” instead of “non-translatable”.

Also, add space throughout the manuscript: e.g., lines 43 and 111 (150 nt, 120 nt, 100 nt).

Line 45: replace “shapes” with “shape”.

Lines 66-67: “The fact that some patterns are region-specific suggests a potential selection by unknown pressure factors.” – This sentence does not add any information. Please consider rewriting or deleting it. Also, consider drift.

Materials and Methods:

Lines 72-73: It is important to mention the criteria used in that search. Mentioning “exhaustive search” is too vague.

Line 74: “redundant sequences were filtered using BLASTn” – Explain better. What parameters were used? E-value, etc.?

Line 75-77: Provide more details about the statistical analysis performed.

Line 77-78: Replace the existing sentence with this one: “Therefore, we obtained a final dataset with 412 complete ZIKV genomes to conduct downstream analyses.”.

Line 93, 210: Flavivirus

Line 92-94: Replace by: “Neighbor-Joining (NJ) phylogenetic tree was performed considering 1000 bootstrap replications and the yellow fever virus (NC_002031.1) as outgroup.”.

Lines 95-98: Use “nucleotide substitution model” instead of “the base model” and “phylogenetic tree” instead of “dendrogram”. Also include how many bootstraps and which is the nucleotide substitution model selected (GTR, JC, etc…). First mention of the algorithm “AIC” in full and then abbreviated.

Consider "phylogenetic tree" throughout the manuscript instead of "dendrograms".

Line 99: Mention the R package and corresponding reference.

Line 115: remove “Block”.

Line 116: allowed.

Line 129: Specify the parameters used to perform the BLASTn.

Results:

Line 153: …to provide… instead of “to have”.

Line 160: countries.

Line 167: It needs a reference.

Consider “clade” instead of “block” throughout the manuscript.

All the Figures: Adjust the readability of all the elements in the figures: scale size, legend, etc. Increase the resolution of all the Figures as well, and provide a comprehensive legend (e.g., “Fan plot type” is not the most relevant information, but some methodological details are – bootstrap replications and models).

Figure 1: Legend – (B) Cladogram showing lineages according to their intra-geographic lineages”. In the legend, the authors should define which are the intra-geographic regions. There is no intra-geographic region for Africa nor for Oceania.

Figure 2A: Why are the authors including a table within Figure 2A with several nucleotide substitution models? What model was used? The authors should consider providing the model used in the legend of the figure, as well as the number of bootstraps.

Figure 2B: In the legend, the authors should define which are the intra-geographic regions. There is no intra-geographic region for Africa nor for Oceania.

Lines 178-180: Please clarify the meaning of “and the resolution distance in the cladogram of continents (Figure 2A) is even smaller than NJ” and provide metrics.

Lines 181-182: Please clarify and provide metrics: “has an even lower resolution than the mathematical approach of distance based on pairwise sequence identities of Figure 1A.”. The algorithms are different – ML calculations use much more complex models than NJ – how can the authors demonstrate “lower resolution”?

Lines 189, 195, 291: “pairwise identity” instead of “paired identity”.

Table 1: Use a decimal point instead of a comma.

Lines 217-221: Consider using (1) or (a) instead of “1.” and start each point using lowercase.

Table 2: Use a decimal point instead of a comma, and the caption comes on top. Use the same number of decimal places.

Table 3: Use a decimal point instead of a comma. Describe the meaning of each element of the first row, as well as of “/” in a footnote below the table.

Lines 258-259: This is not described in the section “Materials and Methods”.

Discussion:

Line 268: “The chronological order” – the authors are not using a molecular clock.

Lines 304-306: Explain what the authors intend to convey with this sentence.

Lines 312-313: typo.

Line 318: 48S … 60S.

Line 325: Aedes africanus

 Lines 335 and 339: Culex spp.

Line 344: in vivo (italic)

Line 361: in-silico (italic)

Line 364: An example of “the RNA 3D structure's relation” and its function can be observed in pre-microRNAs [52].

References:

Lines 404-427: Correct the format (extra spaces and other typos).

Supplementary materials:

Figure S1: Add a “.” after “factor” – “(B) Boxplot of the slip size factor” and remove space from “( p value = 0.0533)”.

Table S1 and Table S2: Use a decimal point instead of a comma and the caption comes on top. Use the same number of decimal places. Correct typos in the caption.

Figure S2: Increase the resolution of the figure.
